# Influence of Spraying Nano-Curcumin and Nano-Glycyrrhizic Acid on Resistance Enhancement and Some Growth Parameters of Soybean (*Glycine max*) in Response to *Tetranychus urticae* Infestation and Drought Stress

**DOI:** 10.3390/plants12010114

**Published:** 2022-12-26

**Authors:** Azza M. Salama, Ahmed M. Ramadan, Hala H. Alakhdar, Thana K. Khan, Hoda A. S. El-Garhy, Tahsin Shoala

**Affiliations:** 1Agricultural Botany Department, Faculty of Agriculture, Cairo University, Giza 12613, Egypt; 2Department of Biological Sciences, Faculty of Science, King Abdulaziz University (KAU), Jeddah 21589, Saudi Arabia; 3Princess Najla bint Saud Al-Saud Center for Excellence Research in Biotechnology, King Abdulaziz University, Jeddah 21589, Saudi Arabia; 4Cotton and Crops Acarology Department, Plant Protection Research Institute, Agricultural Research Centre, Giza 12511, Egypt; 5Genetics and Genetic Engineering Department, Faculty of Agriculture, Benha University, Qalyubia 13736, Egypt; 6Environmental Biotechnology Department, College of Biotechnology, Misr University for Science and Technology, Giza 12568, Egypt

**Keywords:** soybean, nano-curcumin, nano-glycyrrhizic acid, *Tetranychus urticae*, drought, anatomy, growth parameters

## Abstract

Modern nanotechnology has been credited as one of the most significant inventions of the 21st century. Many agricultural disciplines have been affected by nanotechnology in agriculture. Pest control based on natural compounds needs to be enhanced, and enhancing plant growth under climate change conditions, with increasing periods of drought in many countries, is a very vital aim. Thus, the effect of curcumin nanoparticles (Cu-NPs) and glycyrrhizic acid nanoparticles (GA-NPs) as a foliar application under water deficit on natural infestation with the two-spotted spider mite *Tetranychus urticae*, plant growth and yield, anatomical and chemical parameters were investigated during this study. The obtained results revealed that drought stress over the two studied seasons significantly increased the population of *T. urticae* and decreased all morphological and yield characteristics. The application of three mM GA-NPs reduced the mite population average by 39% while using the same concentration of Cu-NPs caused a 33.9% reduction percentage under drought stress. Using 1 mM GA-NPs gave the highest averages of plant height, number of branches, and leaves/plant fresh and dry weight. Moreover, the number of pods, 100 seed weight and seed yield (kg/ha) increased significantly as a result of spraying with GA-NPs under water deficit. From the results, water deficit decreased the values of the leaf and stem anatomical parameters. Treatment with Cu-NPs or GA-NPs under drought stress increased the thickness of mid-vein, xylem, and phloem tissues. Likewise, such treatment increased stem diameter due mainly to the increase in the thickness of cortex, phloem, and xylem tissues compared with the control. Spraying plants with GA-NPs at 1 mM increased the percentages of nitrogen, phosphor, and potassium in seeds in addition to total chlorophyll. Moreover, glutamate, aspartate, leucine, arginine, Lysine, glycine, tyrosine, tryptophan, and methionine concentrations did not differ significantly (*p* > 0.05) in response to all the studied levels of Cu-NPs or GA-NPs either under normal irrigation or drought condition. In light of these findings, researchers and producers should apply and test both Cu-NPs and GA-NP as nano-fertilizer natural sources on economically viable crops.

## 1. Introduction 

Soybean (*Glycine max* (L.)) is a globally significant plant that is widely used to produce a variety of ingredients. It is principally sensitive to water deficiency during the blooming process, the legume, and the seed growing process, whereas it reduces the seed size and yield production. Climate change is projected to have a major impact on precipitation, temperature, and potential evapotranspiration, affecting the frequency and severity of meteorological droughts. One important issue for assessing future impacts is that the effect of changes in meteorological drought has an impact on soil water stress and hydrological droughts such as groundwater and stream water shortages. For instance, soil water drought is relevant to agriculture, land ecosystems, and health by heat waves, whereas the water resources (agriculture, domestic and industrial water use), aquatic, power generation, and sailing of water, among others, have significance in hydrological drought [1].

Drought is a long-term and regionally extensive occurrence of natural water availability below average. Drought is a recurrent global phenomenon with spatial and temporal characteristics which vary considerably across regions. Drought cannot be confused with dryness, which constitutes a long-term average dry climate feature, or water shortages, as long-term imbalances of existing water resources and demand are reflected. However, it is important to note that in arid or semi-arid regions, where the availability of water is already low in normal conditions (aridity), demand is near or farther than natural accessibility, and society can rarely mitigate or adapt to drought [2].

There have been diverse hypotheses about the consequences of drought stress on the development and reproduction of the two-spotted spider mites *Tetranychus urticae* Kosh, (Acari: Tetranychidae). Drought stress increases the number of mites and plant damage, as well as stress in the diet, which affects the density of the spider mites and the egg laid/female population. *T.urticae* is a highly polyphagous and notorious plague that spreads rapidly to develop pesticide resistance [3,4]. It is a major soybean pest, causing a major decrease in yield. The plants present various changes affecting their growth and productivity in stress (biotic or abiotic) conditions. Just a few experiments considered the interactions between a major plague-spider mite and crop waters diet of major plagues-spider mites and crop waters [1,5,6].

Nanotechnology’s application studies the new physical characteristics of different nanosized materials [7]. Nano-sized nanomaterials have changed the world’s view of the same material’s physical structure and size, in addition, have changed the biological research direction towards nanoscience and have become one of the main and most important sciences in various biological aspects such as decreased pre- and post-harvest diseases and environmental pollution [8]. Different materials in their normal size may be toxic, so converting them to nanosize will change their physical properties and increase their toxicity [9]. Converting safe natural products into nanosize could play a dual role in nanoscience by increasing the nano-activity of natural products against various pathogens, increasing fruit and vegetable shelf-life at room temperature, lowering pesticide side-effects and toxicity, treating various human diseases, and reducing the bad or negative side of using unsafe nanomaterials. Depending on your choice of material, nanotechnology application could have a positive or negative effect [10].

Turmeric rhizomes contain most of the curcumin in turmeric. Turmeric (*Curcuma longa* L.) is a rhizomatous herbaceous perennial plant in the Zingiberaceae family. Turmeric is primarily used in traditional Chinese medicine to treat inflammatory conditions; it is also used as a stimulant, anti-bacteria, aspirant, carminative, cordial, astringent, detergent, and diuretic [11,12]. 

Glycyrrhizic acid (GA) is one of the most important bioactive elements in licorice, which is derived from the rhizomes and roots of *Glycyrrhiza glabra*. Licorice has long been used as a herbal medicine to treat a variety of ailments. Glycyrrhizic acid can also be used as Glycyrrhizic acid ammonium salt [13].

This investigation was occurred to study the response of soybean plants to the application of curcumin nanoparticles and Nano-glycyrrhizic acid under drought and normal irrigation conditions, and its relation to the population of the two-spotted spider mite *Tetranychus urticae*.

## 2. Material and Methods

Field experiments were carried out on an experimental farm at Qaha Research Station, Plant Protection Research Institute, Qalyubia governorate, Egypt (30°17′01″ N, 31°12′02″ E) (Figure 1), during the 2020 and 2021 seasons on soybean, (*Glycine max* L. Merrill). Seeds of soybean were secured from Field Crops Institute, Agricultural Research Center, Egypt. Seeds were sown on the first week of May in both seasons. The experimental design was a randomized complete block (RCBD) with four replications. Each plot consisted of six ridges, 70 cm apart and four m long. Seeds were planted at a density of 20 plants per meter of a linear ridge. All other agricultural practices were conducted as recommended for the Qaha station [14].

### 2.1. Synthesis of Nano-Curcumin

Curcumin was purchased from Sigma-Aldrich Company (CAS Number: 458-37-7), Stock of curcumin solution (5 mg/mL) was prepared by dissolving curcumin powder in Dichloromethane (20 mL). One ml of stock solution was added to boiling water (50 mL) in a drop-wise manner under ultra-sonication conditions (XUBA3Analogue Ultrasonic Bath, Grant Company) with an ultrasonic power and frequency of 50 kHz. The solution was sonicated for about 30 min. After sonication, the mixture was stirred at 800 rpm for 20 min till the orange-coloured precipitate was obtained. Thereafter, the supernatant was discarded, and the pellet obtained was used for further study [15].

### 2.2. Glycyrrhizic Acid Ammonium Salt Nanoparticles Preparation

Glycyrrhizic acid ammonium salt purchased from Sigma-Aldrich Company (CAS number: 53956-04-0), 0.1 mg of Glycyrrhizic acid Ammonium salt was dissolved in 1 mL absolute ethanol at room temperature (20–25 °C), then stored in the fridge (4 °C) [16].

### 2.3. Characterization of Nanomaterials by Using Dynamic Light Scattering (DLS)

Measurement of Glycyrrhizic acid (GA-NPs) and Curcumin nanoparticles (Cu-NPs) distribution and size was performed by a dynamic light scattering method using Zetasizer Nano ZS (Malvern Instruments, Malvern, UK) at room temperature. Before measurement, 30 µL of the nanoparticles were diluted with 3 mL of water at 25 °C. Particle size data were expressed as the mean of the Z-average of three independent batches of the nanoparticles.

### 2.4. Treatments

The treatments were controlled at 100% FC; control at 50% FC, curcumin at 0.5 mL/L; curcumin at 1.0 mL/L; glycyrrhizic acid at 0.5 mL/L and glycyrrhizic acid at 1.0 mL/L. A field capacity was calculated under dripping irrigation when zones of water overlap each other on the line. The treatments were applied twice; at 30 and 45 days after sowing. Random samples were taken from each plot 60 days after sowing to record the morphological characters. At harvest (150 days), samples were randomly taken from each pot to determine the yield characteristics. All plants received recommended dose of NPK fertilizers. 

### 2.5. The Seasonal Density of T. urticae on Soybean after Treatment with Cu-NPs and GAS-NPs under Drought and Normal Irrigation

The seasonal density of spider mite, *T. urticae* was determined in plots of different treatments from the last week of May to the first week of October during the two seasons 2020–2021. Ten plant leaves were selected from each replicate fortnightly, and the number of *T. urticae*, was counted with the aid of a stereomicroscope at the Acarology Lab. The mean density of total life stages (immature and adults) was statically analyzed using analysis of variance (ANOVA) and assessed among all treatments of soybean to evaluate the mean reduction percentage and variations in the different treatments and water irrigation mode on the total accumulation number of *T. urticae* [4,17].

### 2.6. The Fertility of T. urticae Females on Treated Soybean Leaves under Drought Stress

Fifteen pots were conducted at Acarology Lab., Plant Protection Research Institute for each irrigation mode and the two watering frequencies (T1—weekly irrigation, T2—two-week irrigation) were applied to soybean as host plants of *T. urticae*. Cu-NPs (3 and 6 mM and GAS-NPs (1 and 3 mM) were sprayed after 30 and 45 days of sowing, three-pot replicates do not spray as a control. Treated leaves were collected from each plant to study the fertility and population buildup of *T. urticae* under laboratory conditions in-vitro conditions at 27 ± 2 °C, 70 ± 5% relative humidity for seven days. In this regard, one female was selected from the stock culture and transferred to a fresh leaf disc of each treated plant. Fresh leaf discs were made which were square or circular, and placed on the cotton bed in a petri-dish plate facing under surface upward. The cotton bed was kept wet by soaking with water twice daily so that the leaf discs remained fresh. Twenty-four hours later, the eggs laid were counted on these leaf discs and removed [18].

### 2.7. Anatomical Studies

The anatomical studies were carried out to investigate the effect of 3 mM Cu-NPs and 1 mM GA-NPs on the leaf and stem anatomy of the soybean plant (stem and leaves) under drought stress (50% FC), represented by the 4th internodes from the plant tip and the leaf was taken from the 4th apical leaf represented by the middle of the lamina including the midrib at the age of 90 days after sowing. Specimens were killed and fixed for at least 48 h in F.A.A. (10 mL formalin, 5 mL glacial acetic acid, 50 mL ethyl alcohol 95%, 35 mL distilled water). Plant materials were washed in 50% ethyl alcohol and dehydrated in a normal butyl alcohol series before being embedded in paraffin wax (melting point 52–54 °C). Transverse sections, 20 µ thick, were cut using a rotary microtome, stained with a double crystal violet/erythrosine combination, and mounted in Canada balsam [19]. Measurements (µm) of the different tissues were taken, and averages of ten readings from five slides were calculated using a micrometer eyepiece and micrometre stage. The slides were microscopically examined and photomicrographed at the Botany Department, Faculty of Agriculture, Cairo University, Egypt. 

### 2.8. Chemical Studies

a.Mineral elements content in leaves

At the vegetation growth stage (60 Days) during the second season leaves from mature plants were subjected to determine total nitrogen (*N*), then calculated crude protein, by multiplying it by 6.25. Phosphorus (*p*), Potassium (K), and total chlorophyll were determined and calculated as % of dry weight according to [20], at the Faculty of Agriculture, Cairo University Research Park (CURP).

b.Determination of amino acids in seeds

At the harvest time during the second season samples from mature dried seeds were subjected to determine amino acids according to the methods described by [21] and measured using an Amino Acid Analyzer (AAA 400 INGOS Ltd., Prague, Czech Republic) at the Faculty of Agriculture, Cairo University Research Park (CURP).

### 2.9. Statistical Analysis

The statistical analysis was carried out using two-way ANOVA using SPSS, ver. 25 (IBM Corp., Armonk, NY, USA, Released 2013). Data were treated as being in a complete randomization design according to Steel et al. (1997) [22]. Multiple comparisons were carried out by applying the Duncun test. The significance level was set at <0.05. 

## 3. Results 

### 3.1. Dynamic Light Scattering (DLS)

The size distribution and stability of prepared curcumin and glycyrrhizic acid nanoparticles were studied using the dynamic light scattering technique, which revealed that the size distribution ranged primarily between 16–30 nm and 5–9 nm, respectively, as shown in (Figure 2 and Figure 4). The zeta potential is an important indicator of colloidal dispersion stability. Colloids with a high zeta potential (a positive value) are thus electrically stabilised (Figure 3 and Figure 5).

### 3.2. The Seasonal Abundance of Spider Mite, Tetranychus Urticae

The soybean’ mode of irrigation had a significant influence on the development of spider mite population density. Plants under water stress conditions were associated with an increase in the number of *T. urticae* mites. This result was observed as early as the first plant exposure to drought. As indicated in Table 1, the mean accumulated number of mites-infested soybean under drought conditions (50% FC) scored the highest percentage 19.76% in comparison with the normal irrigation (100% FC). Additionally, the reduction percentage of mites scored 31%, and 33.9% in response to Cu-NPs 3 mM and Cu-NPs 6 mM treatments respectively under drought conditions. While the percentage of mites’ reduction reached 20.94% and 28.3% in response to Cu-NPs 3 and 6 mM treatments respectively, under normal conditions. The percentage of mites’ reduction scored the highest at 39% under drought conditions compared to 33.11% in normal conditions in response to glycyrrhizic acid GA-NPs 3 mM. All results were recorded during the two studying seasons (Table 1). Obtained results indicated that treated plants with 3 mM of GA-NPs showed a significantly lower number of *T. urticae* than the control in the two seasons using different irrigation methods.

Data in Figure 6A revealed that the average number of insects at a time point (14-AUG) was 224 under drought conditions (50% FC) compared to 175 under normal conditions (100% FC). At time point 14-Aug, the treated plants with 3 and 6 mM of Cu-NPs recorded 190 and 175 insects, respectively, under 50% FC. Exogenous application of Cu-NPs at 3 and 6 mM under 100% FC resulted in 143 and 95 insects, respectively, at time point 14-AUG. Furthermore, the average number of insects at the time point (14-OCT) was 32 under normal conditions (100% FC) compared to 45 insects under drought conditions (50% FC). Interestingly, under normal conditions (100% FC), the plants treated with both studied levels of Cu-NPs recorded 20 and 9 insects, respectively, at time point 14-OCT. However, at the same time point, exogenous application with the two concentrations of Cu-NPs recorded 18 and 9 insects, respectively, under 50% FC.

After GA-NPs treatment, differences in the population density of spider mite, *T. urticae* were observed between the two concentrations (Figure 6B). The average number of insects at a time point (14-AUG) was 224 under drought conditions (50% FC) compared to 160 under normal conditions (100% FC). At time point 14-Aug, the treated plants with 1 mM GA-NPs and 3 mM GA-NPs recorded 165 and 135 insects, respectively, under 50% FC. Exogenous application of 1 mM GA-NPs and 3 mM GA-NPs under 100% FC resulted in 138 and 97 insects, respectively, at time point 14-AUG. Furthermore, the average number of insects at the time point (14-OCT) was 32 under normal conditions (100% FC) compared to 45 insects under drought conditions (50% FC). Interestingly, under normal conditions (100% FC), the treated plants with 1 mM GA-NPs and 3 mM GA-NPs recorded 16 and 2 insects, respectively, at time point 14-OCT. However, at time point 14-OCT, exogenous application of 1 mM GA-NPs and 3 mM GA-NPs recorded 16 and 5 insects, respectively, under 50% FC.

### 3.3. The Fertility of T. urticae Females on Treated Soybean Leaves under Drought Stress

In our experiment, soybean water status affected the abundance of egg-laid per day/mite’s adult females. The average number of mites on plants exposed to water deficit increased significantly throughout the test period, reaching 12 insects under drought conditions (50% FC) compared to 10.3 insects under normal conditions (100% FC) (on the 7th day) (Figure 7A,B). Under normal conditions (100% FC), mite performance was reduced to 6.1 and 5.87 insects in response to 3 mM Cu-NPs and 6 mM Cu-NPs treatments, respectively. Under drought conditions (50% FC), treated plants with 3 mM and 6 mM Cu-NPs scored 9.7 and 7.6 insects, respectively (7th day). While, under normal conditions (100% FC), mite performance was reduced to 6.87 and 5.85 insects in response to 1 mM GA-NPs and 3 mM GA-NPs treatments, respectively. Under drought conditions (50% FC) at time point (7th day), treated plants with 1 mM GA-NPs and 3 mM GA-NPs scored 8.4 and 7.9 insects, respectively (Figure 7B).

### 3.4. Anatomical Studies

#### 3.4.1. Leaf Anatomy

Interestingly, treatments with Cu-NPs and GA-NPs enhanced the anatomical characteristics of leaves (Table 2 and Figure 8). Under drought conditions (50% FC), anatomical characteristics of leaves enhanced significantly in response to 1 mM GA-NPs. The recorded data were, 1125 µm midvein thick, 160 µm Lamina, 65 µm plaside thickness, 210 µm Xylem tissue thick, 155 µm Phloem tissue thick, and dimensions of the main vascular bundle of mid-vein (540 µm length −770 µm width). However in 3 mM Cu-NPs, leaf midvein was 945 µm, Lamina was 184 µm, palisade thickness was 74 µm, Xylem tissue was 195 µm, Phloem tissue (150 µm), and Dimensions of the main vascular bundle of mid-vein were 445 µm length and 485 µm width. Wihle control 50% FC data were 905 µm, 175 µm, 70 µm, 160 µm, 110 µm and 265 µm length −320 µm width, respectively. Control 100% FC data were 930 µm, 165 µm, 68 µm, 177 µm, 135 µm and 385 µm length −405 µm width, respectively.

#### 3.4.2. Stem Anatomy

Under drought conditions (50% FC), anatomical characteristics of the stem enhanced significantly in response to 1 mM GA-NPs and recorded 3287 µm Main stem diameter, 220 µm Cortex thickness, 22 vascular bundles, 240 µm phloem tissue thickness, 650 µm Xylem tissue thick and 1685 µm Parenchymatous pith thick respectively, compared to control (50% FC–100% FC). Correspondingly, the treated plants with 3 mM CU-NPs enhanced anatomical characteristics of stem and scored 3177µm Main stem diameter, 165 µm Cortex thickness, 18 vascular bundles, 215 µm phloem tissue thickness, 590 µm Xylem tissue thick and 1620 µm Parenchymatous pith thick respectively, compared to control (50% FC–100% FC) (Table 3).

Results presented in Table 3 and Figure 9 indicated that foliar spray with 3 mM Cu-NPs increased the diameter of the main stem by 45.1%. The number of vascular bundles was increased by 20.0%. The increased thickness of phloem and xylem tissues and parenchymatous pith were 65.4, 131.4, and 21.8 %, respectively. While the thickness of the cortex decreased by 15.4% less than the control. Moreover, the thickness of the main stem increased with the application of GA-NPs at 1 mM over control by 50.1%. The number of vascular bundles increased by 46.7 %. On the other hand, the thickness of cortex, phloem, xylem tissues, and parenchymatous pith was markedly increased under the same conditions by 12.8, 84.6, 154.9, and 26.7% respectively. On the other hand, water deficit (50% FC) caused significant decreases in all anatomical characters except cortex thickness which was increased by 14.7% and the number of vascular bundles was increased by 25% compared with the control. Whereas the main stem diameter decreased by 27.5% below control. The number of vascular bundles was increased by 25% compared with the control. While the thickness of phloem and xylem tissues and parenchymatous pith decreased by 13.3, 20.3, and 14.5% respectively, compared with the control (Table 3).

### 3.5. Growth Characters

#### 3.5.1. Effect of Drought

Results in Table 4 revealed that drought stress (50% FC) caused significant decreases for studied morphological characters compared to normal irrigation (100% FC). Plant height at 50% FC decreased significantly by 10.1% below plants irrigated with 100% FC. As to branches and leaves, numbers recorded a decrease in plants with 50% FC by 22.2 and 19.4% compared to control. The same trend was observed with a fresh and dry weight of shoot, plants irrigated with 100% FC recorded the highest values, being 31.0 and 18.6% more than plants irrigated with 50% FC for plant fresh and dry weight, respectively.

#### 3.5.2. Effect of Nano-Curcumin (Cu-NPs)

Data recorded in Table 4 indicated that all studied growth parameters of soybean; plant height, number of branches and leaves/plant, fresh and dry weights/plant were significantly affected by foliar spray with Cu-NPs under water deficit in both seasons compared with untreated plants. Cu-NPs at 6 mM were the most effective treatment in increasing growth parameters in both seasons. The maximum plant height of 88.27 cm was achieved at a concentration of 6 mM nano-curcumin, being 71.3% more than the control (50% FC). Concerning the number of branches, it increased significantly with increasing the Cu-NPs concentration. Branch numbers were 10 at 3 mM, being 42.8% more than the control, while 6 mM recorded 11 being 57.1% over the control. On the other hand, leaves number at 6 mM exhibited 69.5 compared to 29.0 (control), which is 139.6% more than the control. As to the effect on fresh weight/plant, there was a significant increase in fresh weight with increasing the concentration. Data proved that the plant recorded 140.1 and 143.0 g, being 140.0 and 145.3% more in comparison with the control for 3 and 6 mM, respectively. Regarding the effect on dry weight/plant, it is clear from Table 4 that, spraying plants with 6 mM recorded 63.4 g compared to 19.3 g for control, being 228.5% over control.

#### 3.5.3. Effect of Nano-Glycyrrhizic Acid (GA-NPs)

As shown in Table 4 two concentrations with GA-NPs significantly increased all investigated morphological characters in both studied seasons compared with the control. The most increasing vegetation growth was observed when plants were treated with 3 mM of GA-NPs in both seasons. The highest concentration (3 mM) produced the tallest plants (88.5 cm), 71.8% more than the control. Table 4 showed that plants treated with 1 mM recorded nine branches, 28.6% more than the control whereas; a concentration of 3 mM recorded twelve branches, being 71.4% over the control. On the other hand, the leaves number exhibited the highest number at 3 mM being 148.3% over control. The maximum fresh weight of the plant 147.17 g was obtained at 3 mM, 155.7% more than the control. The same trend was obtained with plant dry weight, at 1 mM exhibited 41.8 g, being 116.6% more than the control Table 4, while at 1 mL/L recorded the highest value 69.1 g, being 258.0% more than the control.

### 3.6. Yield and Its Contents

Table 5 revealed that water defect (50% FC) decreased the number of pods/plants by 50.7% in comparison with the control. On the other hand, the weight of 100 seeds decreased below control by 2.2%. Finally, the seed and total yield of the plant decreased because of water defects at 47.3 and 48.8% below control. Relative to the control, plants showed a gradual increase in pod numbers as curcumin concentration increased. Which increased by 163.0 and 175% at 0.5 and 1.0 mL/L, respectively. The weight of 100 seeds increased over control by 9.6 and 15.5% at the same previous concentrations, respectively. Spray Cu-NPs at 3 mM increased seed and total yield by 26.6 and 29.3% over control. As to the effect of GA-NPs, the number of pods/plants exhibited the highest at 3 mM in comparison with control by 210%. Spraying GA-NPs at 3 mM recorded the highest weight of 100 seeds by 20.7% over control. Seed and total yield increased by 42.7 and 43.9% over control because of spray GA-NPs at 3 mM.

### 3.7. Mineral Elements Content in Leaves

Data in Table 6 illustrated the effect of the foliar application with two concentrations of Cu-NPs and GA-NPs on the percentage of nitrogen, phosphor, potassium, and total chlorophyll in dry leaves under drought stress and natural infestation of spider mite, *T. urticae*. It is obvious from leaves analysis that, there are differences between the two studied materials where curcumin nanoparticles at 3 mM recorded an increase by 37.1, 31.0, and 26.8 for the percentage of *N*, *p*, and K, respectively more than control, while spraying at 6 mM recorded an increase by 39.9, 44.8, and 31.4% respectively more than the control for the previous elements. As to total chlorophyll percentage, treatment with 1.0 mL/L gave the highest increase percentage (71.7%) more than untreated plants (control). Foliar application of nano-glycyrrhizic acid at 3 mM significantly increased the percentage of *N*, *p*, and K by 61.5, 65.5, and 39.4% respectively. Concerning the effect of nano-glycyrrhizic acid on the percentage of total chlorophyll, spray with 1 mM recorded the highest percentage (85.1%) more than the control (Table 6).

### 3.8. Chemical Studies

#### Determination of Amino Acids in Seeds

Analysis of amino acids in soybean seeds detected ten types (Table 7). These types were divided into essential amino acids i.e., Methionine, Tryptophan, Leucine, and Lysine, and nonessential amino acids, i.e., Glutamate, Glycine, Alanine, Arginine, Tyrosine, and Aspartate. Seeds of soybean contain a high concentration of Glutamate, a moderate concentration of Aspartate, Leucine, Arginine, Lysine, Glycine, Alanine, and Tyrosine, and a low concentration of Methionine and Tryptophan. Regarding the effect of treatment with two concentrations of nano-curcumin or nano-glycyrrhizic acid on the content of the amino acid of soybean seeds, the obtained results showed that foliar spraying nano-glycyrrhizic acid at 3 mM caused a marked increase in the concentration of all essential and nonessential amino acids comparing with the control plants. The increased percentage in the content of the amino acid recorded 4.5% in Glutamate, 21.4% in Glycine, 37.5% in Methionine, 18.5% in Alanine, 12.4% in Arginine, 28.6% in Tyrosine, 24.6% in Tryptophan, 13.6% in Leucine, 7.8% in Lysine, and 4.7% in Aspartate compared with untreated plants.

## 4. Discussion

Nanotechnology has been considered one of the most important technologies of the modern Era. Nanotechnology has been applied in agriculture with great impact in many agricultural disciplines [9]. For instance, natural nanomaterials can enhance plant growth, secondary metabolites, active compounds, resistance to various stimuli, crop yields, and distinct physiological pathways [23].

In this study, it is logical that we found the *T. urticae* attacking the drought-stressed plant slightly more than the relaxed plant; in normal conditions, even the difference is not significant. However, the positive impact of nano-glycyrrhizic acid and nano-curcumin treatments is a reduction in the no. of insects, significantly in both drought and normal conditions, in both concentrations of Cu-NPs and GA-NPs (Table 1, Figure 5). It may be the increase in resistance is correlated with increases in nitrogen, potassium, and phosphorus content, as well as an increase in photosynthetic pigments and capacity (Table 2 and Table 6), which is unfavourable signalling for *T. urticae* [24,25] and direct causes an observed reduction in its populations. Current research studies proved that the application of nano-glycyrrhizic acid and nano-curcumin on the soybean (*Glycine max*) enhanced the photosynthesis rate by increasing Leaf area root length, water, and nutrient materials absorption, which impacted chlorophyll percentage, plant height, number of branches, fresh and dry weight, oil percentage and contents, and plant resistance against both biotic and abiotic stresses (Table 2 and Table 3 and Figure 6). The impact of nanoparticle treatments in plant yields was reported before [26] when the investigator clarified that a better growth of *Solanum lycopersicum* and high fruit yield due to TiO_2_ NPs foliar treatment. Also, the highly favourable effects of Ag NPs on grains/spike, 100-grain weight, and grain yield per pot in wheat [27]. Exogenous application of nano-glycyrrhizic acid and nano-curcumin to the soybean (*Glycine max*) may induce specific pathways that have a significant impact on plant growth, physiological activities, secondary metabolites, various pathways, and active compound production [4,23].

Our findings show that nano-glycyrrhizic acid and nano-curcumin improve plant growth by increasing photosynthesis pathways and growth hormones (Table 2 and Table 3). Many investigators explain that this increase in plant growth is due to the effect of nano-glycyrrhizic acid and nano-curcumin MAPKs pathway induction which regulates many important cellular processes such as cell division, different developmental processes regulated by hormones, stress responses, metabolism, and biologically active compounds [28] in addition to the role of MAPK cascades in stomata controlling, callose deposition, and seed germination [29,30]. Other investigators suggest the positive effect of nano-glycyrrhizic acid and nano-curcumin on the Oxidative signal Inducible1 gene (Oxi1) a protein kinase member of the AGC family that links ROS accumulation to plant response and resistance to various stimuli, as well as oxidative burst-mediated signaling in plant roots [31,32].

In leaves, that exogenous application with 3 mM Cu-NPs increased the thickness of midvein and lamina by 4.4 and 5.1% respectively, when compared with the control (50% FC). The increase in lamina thickness was accompanied by increments in the thickness of palisade and spongy tissues by 5.7 and 16.7% at 3 mM above the control, respectively. Likewise, the main vascular bundle of the midvein was increased in size as a result of spray with Cu-NPs- 3 mM. The increase was mainly due to the increase in the xylem by 21.9% and phloem by 36.4% more than the control. In stem, 3 mM Cu-NPs treatment increased the diameter of the main stem by 45.1%, the number of vascular bundles was increased by 20.0 %, the increased thickness of phloem and xylem tissues, and parenchymatous pith were 65.4, 131.4, and 21.8%, respectively. While the thickness of the cortex decreased by 15.4. Moreover, the thickness of the main stem increased with the application of GA-NPs at 1 mM. The number of vascular bundles increased by 46.7%. On the other hand, the thickness of cortex, phloem, xylem tissues, and parenchymatous pith was markedly increased under the same conditions by 12.8, 84.6, 154.9, and 26.7% respectively.

Concerning the effect of GA-NPs acid on leaf structure, it was indicated that the thickness of midvein, xylem, and phloem tissues increased markedly at 1 mM over the untreated plants by 24.3, 31.2, and 40.9%, respectively. Additionally, the thickness of lamina, palisade, and spongy tissues decreased by 8.6, 7.1, and 16.7%, respectively at 1 mM as compared to non-treated plants. The size of the midrib was further increased, as the percentages increased by 42.9% in length and 87.3% in width, referring to an increase in the dimensions of the vascular bunch of the midrib since it reached 50% of the length and 200% of the width, along with the thickness of the collenchyma above the bundle (40%) and below the vascular bundle (52.5%). The increase in midrib vascular bundle dimensions is due to 80% growth of the thickness of phloem tissue and 71.4% of xylem tissue.

On leaves with a thicker mesophyll rose, the lowest density of phytophagous mites occurred. Some species were more resistant to spider mites with thicker cuticles and epidermis. Furthermore, mites favored less the upper leaf surface, as they are thicker cuticles and epidermis than the lower surface [14,33]. Within most often feed exclusively on cells within the mesophyll parenchyma; the disturbed cells adjacent to the leaf surface’s epidermal layer from which the mites feed. As the thickness of leaves ranges from 100 to 150 μm, the style of the mites can completely cross a leaf and it can be reached by a palisade or by spongy mesophyll regardless of the surface of the leaf; however, the basis is currently unknown for this preference [34]. Although the treatment with nano-glycyrrhizic acid gives positive results in most traits, we can say the nano-glycyrrhizic acid and nano-curcumin treatments significantly improved most soybean traits and tolerance to biotic and abiotic stress.

## 5. Conclusions

Natural nanomaterials such as curcumin and glycyrrhizic acid, in both concentrations, improved soybean growth parameters. Curcumin nanoparticles and glycyrrhizic acid nanoparticles may both act as nano-fertilizer sources, mitigating the negative effects of water scarcity on soybean growth and yield. These treatments resulted in a significant decrease in *T. urticae* densities over the growing season, as well as a decrease in mite female fertility. The highest concentrations of curcumin and glycyrrhizic acid nanoparticles had nearly the same impact as lower concentrations, so low concentrations were recommended for economic impact applications. Both curcumin nanoparticles and glycyrrhizic acid ammonium salt nanoparticles may act as nano-fertilizer sources to alleviate the harmful effect of water deficiency on soybean growth and its yield. In light of these findings, researchers and producers should apply and test nano-fertilizers on economically viable crops.

## Figures and Tables

**Figure 1 plants-12-00114-f001:**
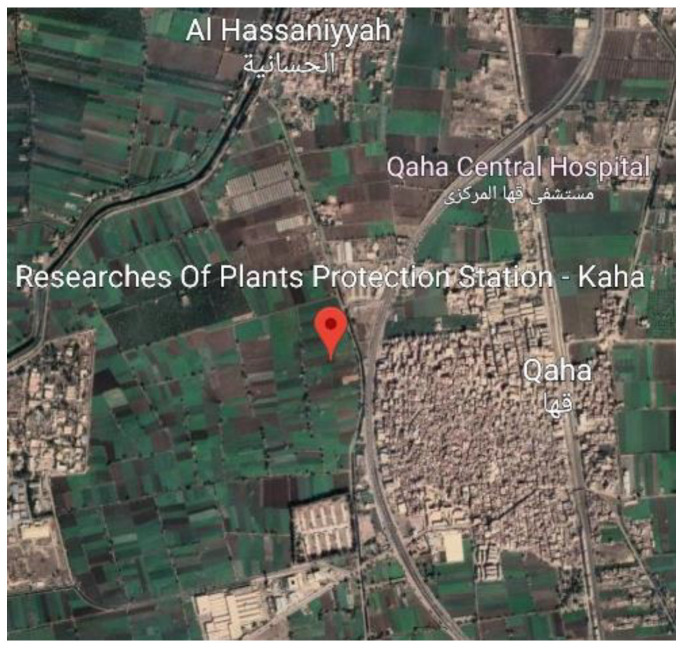
Location of Research of Plant Protection Station, Qaha, Qalyubia, Egypt (30°17′01″ N, 31°12′02″ E).

**Figure 2 plants-12-00114-f002:**
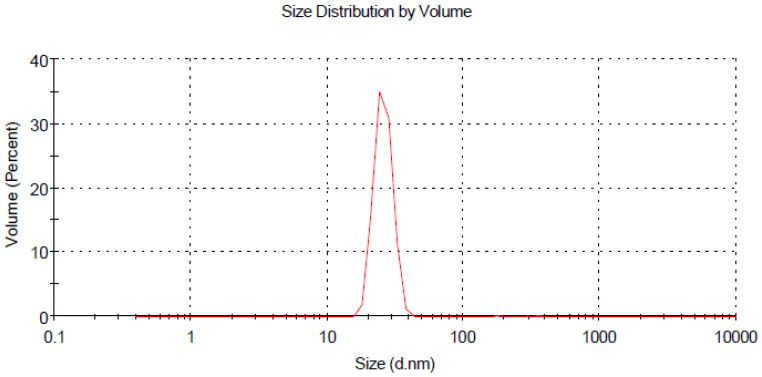
Z average size of curcumin nanoparticles.

**Figure 3 plants-12-00114-f003:**
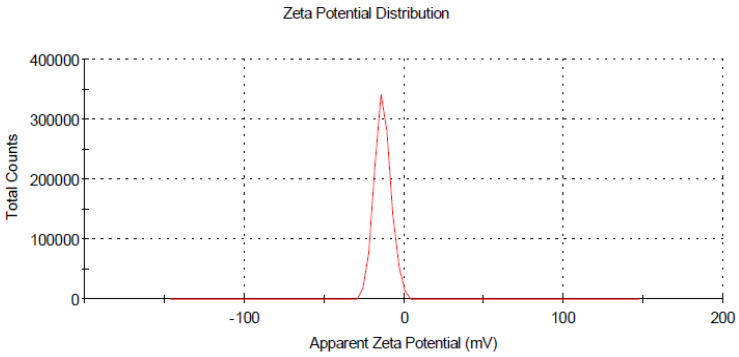
Zeta potential for curcumin nanoparticles.

**Figure 4 plants-12-00114-f004:**
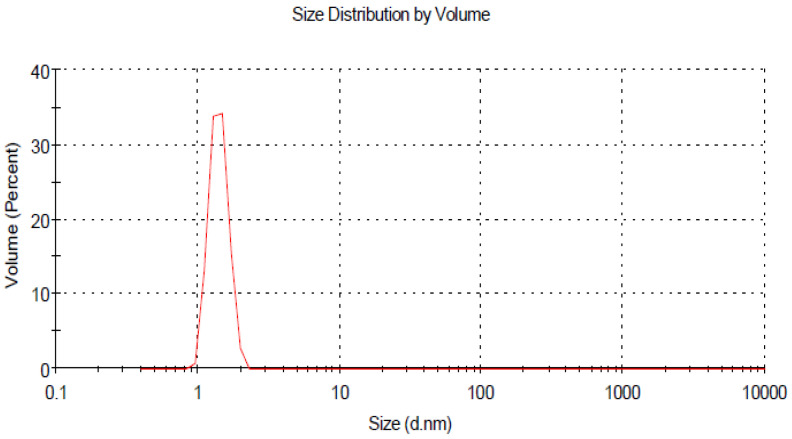
Z average size of Glycyrrhizic acid nanoparticles.

**Figure 5 plants-12-00114-f005:**
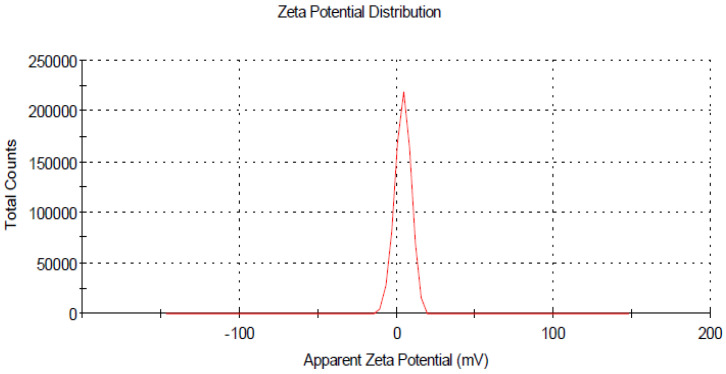
Zeta potential for Glycyrrhizic acid nanoparticles.

**Figure 6 plants-12-00114-f006:**
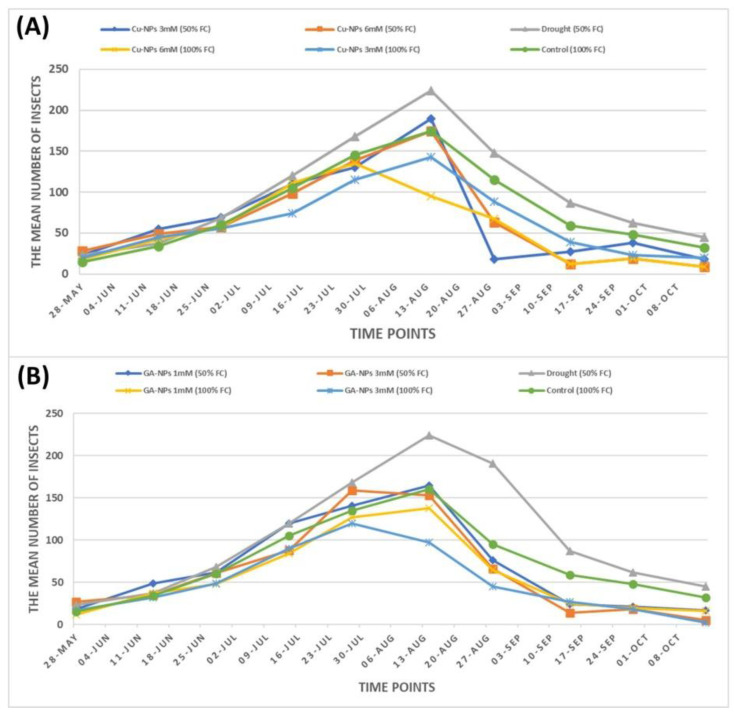
(**A**). Mean of seasonal abundance of mites infesting soybean plants under treatments by Cu-NPs at 3 and 6 mM, (**B**). Mean of seasonal abundance of mites infesting soybean plants under treatments by GA-NPs at 1 and 3 mM under normal irrigation (100% FC) and drought stress (50% FC), during the successive growing seasons of 2020–2021.

**Figure 7 plants-12-00114-f007:**
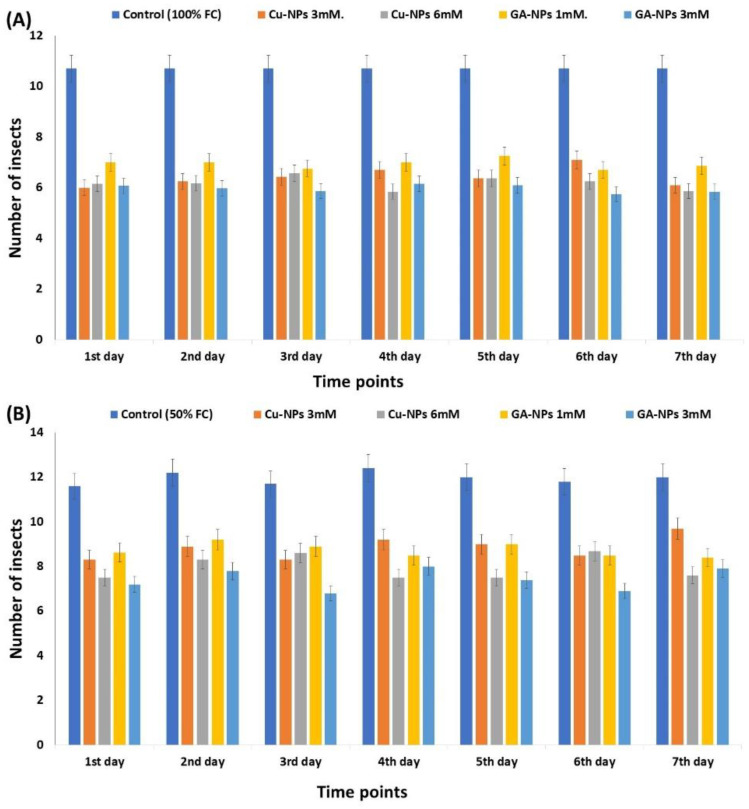
(**A**). The effect of Cu-NPs and GA-NPs at different concentrations on the number of *T. urticae* at different time points in response to normal watering treatment (100% FC), (**B**). The effect of Cu-NPs and GA-NPs at different concentrations on the number of *T. urticae* at different time points in response to drought treatment (50% FC).

**Figure 8 plants-12-00114-f008:**
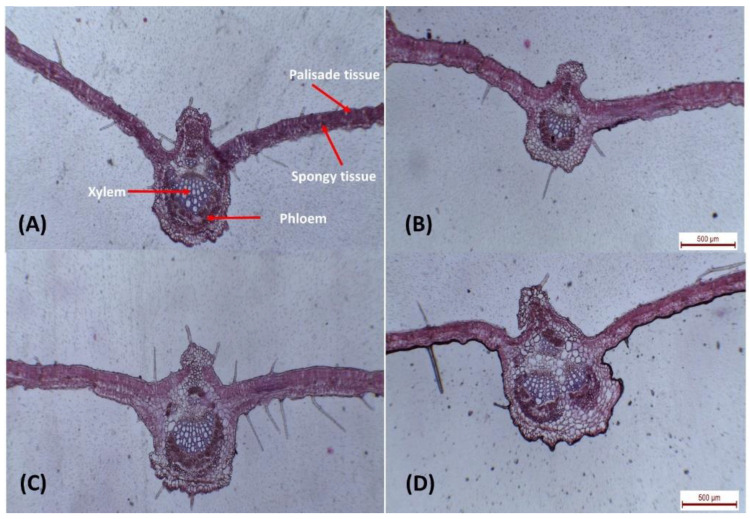
Transverse sections through the blade of soybean leaves at the age of 90 days, (**A**). Control plant (100% FC), (**B**). Negative control plant (50% FC), (**C**). Plants sprayed with Cu-NPs at 3 mM, and (**D**). Plant sprayed with GA NPs at 1 mM, 40×.

**Figure 9 plants-12-00114-f009:**
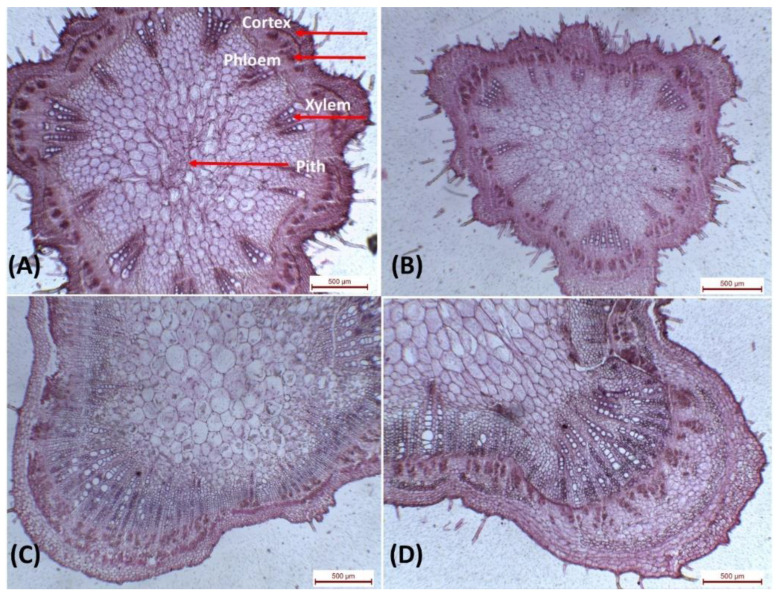
Transverse sections through the 4th internode of soybean stem at the age of 90 days, (**A**): Control plants (100% FC), (**B**): Control plants (50% FC), (**C**): Plants sprayed with Cu-NPs at 3 mM, and (**D**): Plants sprayed with GA-NPs at 1 mM, 100×.

**Table 1 plants-12-00114-t001:** Mean accumulative number of *T. urticae* on soybean plants in response to Cu-NPs and GA-NPs during the successive growing seasons of 2020–2021 under normal and drought stress.

Drought	Control	Cu-NPs 3 mM	Cu-NPs 6 mM	GAS-NPs 1 mM	GAS-NPs 3 mM
50% FC	982 ± 9.79 ^a^	678 ± 4.9 ^c^	649 ± 7.35 ^d^	693 ± 13.47 ^b^	626 ± 20.41 ^e^
Reduction %	-	31%	33.9%	32.43%	39%
100% FC	788 ± 14 ^a^	623 ± 9.39 ^b^	565 ± 12.66 ^c^	572 ± 12.25 ^c^	497 ± 8.16 ^d^
Reduction %	19.76%	20.94%	28.3%	23.02%	33.11%

a, b, c, d and e: There is no significant difference (*p* > 0.05) between any two means, within the same raw have the same superscript letter. The data represent the means ± standard error.

**Table 2 plants-12-00114-t002:** The effect of Cu-NPs (3 mM) and GAS-NPs (1 mM) on anatomical characters of soybean leaf (µm) under drought stress (50% FC).

Treatments	Midvein Thick	Lamina Thick	Palisade Thick	Spongy Thick	Dimensions of the Main Vascular Bundle of Mid-Vein		Xylem Tissue Thick	Phloem Tissue Thick
					Length	Width		
Control (100% FC)	930 ± 41.84 ^b^	165 ± 2.89 ^b^	68 ± 0.95 ^ab^	82 ± 6.13 ^bc^	385 ± 2.75 ^c^	405 ± 24.66 ^c^	177 ± 8.94 ^bc^	135 ± 2.00 ^b^
Drought (50% FC)	905 ± 7.64 ^b^	175 ± 3.59 ^abc^	70 ± 2.04 ^ab^	90 ± 1.64 ^b^	265 ± 8.24 ^d^	320 ± 7.67 ^d^	160 ± 6.87 ^c^	110 ± 6.47 ^c^
Cu-NPs (3 mM) + 50% FC	945 ± 9.28 ^b^	184 ± 4.10 ^a^	74 ± 2.81 ^a^	105 ± 3.14 ^a^	445 ± 8.75 ^b^	485 ± 10.77 ^b^	195 ± 3.24 ^ab^	150 ± 5.13 ^a^
GA-NPs (1 mM) + 50% FC	1125 ± 11.64 ^a^	160 ± 5.27 ^b^	65 ± 3.06 ^b^	75 ± 0.61 ^c^	540 ± 10.12 ^a^	770 ± 5.03 ^a^	210 ± 4.04 ^a^	155 ± 2.84 ^a^

The data represent the means ± standard error. Means followed by the same superscript letter in each column were not significantly different according to Duncan’s multiple range test (*p* ≤ 0.05).

**Table 3 plants-12-00114-t003:** The effect of Cu-NPs (3 mM) and GAS-NPs (1 mM) on anatomical characters of soybean stem (µm) under drought stress (50% FC).

Treatments	Main Stem Diameter	Cortex Thickness	Number of Vascular Bundles	Phloem Tissue Thickness	Xylem Tissue Thick	Parenchymatous Pith Thick
Control (100% FC)	3020 ± 38.14 ^b^	170 ± 5.03 ^b^	12 ± 0.40 ^d^	150 ± 1.00 ^c^	320 ± 1.15 ^c^	1555 ± 7.23 ^c^
Drought (50% FC)	2190 ± 51.52 ^c^	195 ± 2.13 ^a^	15 ± 0.38 ^c^	130 ± 1.15 ^d^	255 ± 1.53 ^d^	1330 ± 4.51 ^d^
Cu-NPs (3 mM) + 50% FC	3177 ± 61.42 ^ab^	165 ± 4.41 ^c^	18 ± 0.40 ^b^	215 ± 3.06 ^b^	590 ± 7.37 ^b^	1620 ± 4.73 ^b^
GA-NPs (1 mM) + 50% FC	3287 ± 43.40 ^a^	220 ± 3.27 ^a^	22 ± 0.70 ^a^	240 ± 1.15 ^a^	650 ± 7.55 ^a^	1685 ± 4.36 ^a^

The data represent the means ± standard error. Means followed by the same superscript letter in each column were not significantly different according to Duncan’s multiple range test (*p* ≤ 0.05).

**Table 4 plants-12-00114-t004:** Effect of irrigation treatments, Cu-NPs and GA-NPs applications, and its interaction on growth parameters of soybean.

Parameters	Irrigation	Nutrition Applications				
		Control	Cu-NPs (3 mM)	Cu-NPs (6 mM)	GA-NPs (1 mM)	GA-NPs (3 mM)
Plant Height (cm)	Normal (100% FC)	56.43 ± 1.05 ^aD^	79.93 ± 0.43 ^bB^	81.43 ± 0.34 ^bAB^	74.47 ± 1.87 ^bC^	83.07 ± 1.49 ^bA^
	Drought (50% FC)	51.20 ± 0.94 ^bD^	85.93 ± 1.04 ^aB^	88.27 ± 0.82 ^aA^	79.37 ± 0.75 ^aC^	88.50 ± 1.00 ^aA^
	Mean	53.82 ± 1.33 ^D^	83.10 ± 1.36 ^B^	84.85 ± 1.58 ^AB^	76.92 ± 1.42 ^C^	85.79 ± 1.46 ^A^
No. of branches/Plant	Normal (100% FC)	8.67 ± 0.88 ^aB^	10.00 ± 1.00 ^aAB^	9.67 ± 0.33 ^bAB^	8.66 ± 0.67 ^aB^	10.67 ± 0.33 ^bA^
	Drought (50% FC)	6.67 ± 0.33 ^bD^	10.00 ± 0.58 ^aBC^	10.67 ± 0.67 ^aB^	8.66 ± 0.33 ^aC^	12.00 ± 1.00 ^aA^
	Mean	7.67 ± 0.61 ^C^	10.00 ± 0.52 ^AB^	10.17 ± 0.40 ^A^	8.67 ± 0.33 ^BC^	11.33 ± 0.56 ^A^
No. of leaves/plant	Normal (100% FC)	36.00 ± 1.53 ^aD^	55.00 ± 2.52 ^bB^	59.00 ± 1.53 ^bAB^	47.33 ± 2.60 ^bC^	63.00 ± 2.65 ^bA^
	Drought (50% FC)	29.00 ± 1.53 ^bD^	59.00 ± 2.65 ^aB^	69.33 ± 2.85 ^aA^	53.33 ± 1.2 ^aC^	72.00 ± 4.04 ^aA^
	Mean	32.50 ± 1.84 ^D^	57.00 ± 1.86 ^B^	64.17 ± 2.73 ^A^	50.33 ± 1.86 ^C^	67.50 ± 2.95 ^A^
Shoot F.W (g)	Normal (100% FC)	84.50 ± 1.15 ^aE^	134.33 ± 2.85 ^bC^	140.00 ± 2.52 ^bB^	123.96 ± 1.00 ^aD^	145.33 ± 1.76 ^aA^
	Drought (50% FC)	58.30 ± 0.57 ^bD^	141.47 ± 1.69 ^aB^	143.83 ± 1.66 ^aAB^	102.53 ± 1.12 ^bC^	147.17 ± 2.33 ^aA^
	Mean	71.40 ± 5.89 ^E^	137.90 ± 2.18 ^C^	141.92 ± 1.6 B	113.25 ± 4.84 ^D^	146.25 ± 1.37 ^A^
Shoot D.W (g)	Normal (100% FC)	23.37 ± 0.73 ^aE^	53.33 ± 1.38 ^bC^	56.50 ± 0.95 ^bB^	44.87 ± 0.84 ^aD^	60.00 ± 2.65 ^bA^
	Drought (50% FC)	19.83 ± 0.38 ^bE^	58.27 ± 1.47 ^aC^	63.47 ± 1.32 ^aB^	42.00 ± 0.95 ^bD^	69.70 ± 1.40 ^aA^
	Mean	21.60 ± 0.87 ^E^	55.80 ± 1.43 ^C^	59.99 ± 1.72 ^B^	43.44 ± 0.86 ^D^	64.85 ± 2.55 ^A^

a, b & c: There is no significant difference (*p* > 0.05, LSD) between any two means for each attribute, within the same column have the same superscript letter; A, B, C, D & E: There is no significant difference (*p* > 0.05, LSD) between any two means for each attribute, within the same row have the same superscript letter. The data represent the means ± standard error.

**Table 5 plants-12-00114-t005:** Effect of irrigation treatments, Cu-NPs, and GA-NPs applications, and its interaction on the yield of soybean.

Parameters	Irrigation	Nutrition Applications				
		Control	Cu-NPs (3 mM)	Cu-NPs (6 mM)	GA-NPs (1 mM)	GA-NPs (3 mM)
Number of pods/plants	Normal (100% FC)	138.00 ± 1.00 ^bD^	145.00 ± 1.53 ^bC^	170.00 ± 1.53 ^bB^	139.00 ± 1.53 ^bD^	178.00 ± 1.53 ^bA^
	Drought (50% FC)	168.00 ± 1.53 ^aD^	179.00 ± 1.53 ^aC^	187.00 ± 1.53 ^aB^	161.00 ± 1.53 ^aE^	211.00 ± 3.06 ^aA^
	Mean	153.00 ± 6.76 ^D^	162.00 ± 7.66 ^C^	179.00 ± 3.92 ^B^	150.00 ± 5.01 ^D^	195.00 ± 7.54 ^A^
100 seed weight (g.)	Normal (100% FC)	13.87 ± 0.12 ^aB^	13.67 ± 0.29 ^bB^	15.13 ± 0.42 ^bA^	13.60 ± 0.13 ^bB^	15.47 ± 0.28 ^bA^
	Drought (50% FC)	13.50 ± 0.12 ^bC^	14.93 ± 0.07 ^aB^	15.80 ± 0.08 ^aA^	13.90 ± 0.16 ^aC^	15.79 ± 0.13 ^aA^
	Mean	13.69 ± 0.11 ^C^	14.30 ± 0.31 ^B^	15.47 ± 0.24 ^A^	13.75 ± 0.11 ^C^	15.63 ± 0.16 ^A^
Seed yield/plant (g.)	Normal (100% FC)	41.93 ± 0.35 ^aD^	49.00 ± 0.57 ^bC^	51.60 ± 0.14 ^aB^	41.97 ± 0.19 ^bD^	57.00 ± 0.60 ^bA^
	Drought (50% FC)	22.00 ± 0.78 ^bE^	49.87 ± 0.53 ^aC^	51.80 ± 0.12 ^aB^	44.86 ± 0.14 ^aD^	58.93 ± 0.64 ^aA^
	Mean	31.97 ± 4.47 ^E^	49.44 ± 0.40 ^C^	51.70 ± 0.09 ^B^	43.42 ± 0.65 ^D^	57.97 ± 0.58 ^A^
Plant yield t/h	Normal (100% FC)	0.41 ± 0.01 ^aD^	0.44 ± 0.01 ^bC^	0.46 ± 0.01 ^bB^	0.39 ± 0.01 ^bE^	0.53 ± 0.01 ^bA^
	Drought (50% FC)	0.21 ± 0.01 ^bD^	0.50 ± 0.02 ^aB^	0.52 ± 0.01 ^aB^	0.45 ± 0.01 ^aC^	0.59 ± 0.02 ^aA^
	Mean	0.31 ± 0.04 ^D^	0.47 ± 0.02 ^B^	0.49 ± 0.01 ^B^	0.42 ± 0.01 ^C^	0.56 ± 0.02 ^A^

a & b: There is no significant difference (*p* > 0.05, LSD) between any two means for each attribute, within the same column have the same superscript letter; A, B, C, D & E: There is no significant difference (*p* > 0.05, LSD) between any two means for each attribute, within the same row have the same superscript letter. The data represent the means ± standard error.

**Table 6 plants-12-00114-t006:** The effect of Cu-NPs and GA-NPs on mineral element content (%matter) in soybean leaves under drought stress. Each value is the average of both seasons.

Treatments		*N* %	*p* %	K %	Total Chlorophyll	Crude Protein
Control (100% FC)		2.95	0.19	1.89	3.31	18.43
Drought (50% FC)		2.86	0.29	1.75	2.69	17.87
Cu-NPs + 50% FC	(3 mM)	3.92	0.38	2.22	4.11	24.50
	(6 mM)	4.00	0.42	2.30	4.62	25.00
GA-NPs+ 50% FC	(1 mM)	3.81	0.37	1.93	4.09	23.81
	(3 mM)	4.62	0.48	2.44	4.98	28.87

**Table 7 plants-12-00114-t007:** Effect of Cu-NPs and GA-NPs on distribution of amino acids (% of dry matter) in soybean seeds under drought stress (50% FC). Each value is the average of both seasons.

Treatments		Glutamate	Glycine	Methionine	Alanine	Arginine	Tyrosine	Tryptophan	Leucine	Lysine	Aspartate
Control (100% FC)		79.6	19.5	5.7	16.8	27.6	13.4	5.6	32.2	23.5	40.8
Drought (50% FC)		77.8	18.2	4.8	16.2	25.8	11.9	5.7	30.8	23.1	40.2
Cu-NPs + 50% FC	3 mM	80.7	20.6	6.4	18.0	28.5	14.8	6.4	33.4	24.5	41.6
	6 mM	80.8	21.5	6.8	16.5	26.7	15.5	6.5	32.4	22.5	40.6
GA-NPs + 50% FC	1 mM	80.5	20.3	6.0	17.1	28.4	14.5	6.1	33.0	24.1	41.5
	3 mM	81.3	22.1	6.6	19.2	29.0	15.3	7.1	35.0	24.9	42.1

## Data Availability

The data presented in this study are available upon request from the corresponding author. The data are not publicly available due to privacy concerns.

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
