# Peer review of "Influence of Spraying Nano-Curcumin and Nano-Glycyrrhizic Acid on Resistance Enhancement and Some Growth Parameters of Soybean (Glycine max) in Response to Tetranychus urticae Infestation and Drought Stress"

_plants, 2022, doi:10.3390/plants12010114_

Round 1

Reviewer 1 Report

Dear Authors,

I have an opportunity to review manuscript entitled : “ Influence of spraying Nano-curcumin and Nano- glycyrrhizic acid on resistance enhancement and some growth parameters of soybean (Glycine max) in response to Tetranychus urticae infestation and drought stress “ submitted to the Plants MDPI Journal.

Authors concentrated on the effect of curcumin nanoparticles (Cu-NPs) and glycyrrhizic acid ammonium salt (GAS-NPs) as a foliar application under water deficit on natural infestation with the two-spotted spider mite Tetranychus urticae, plant growth, yield, anatomical and chemical parameters.

Interestingly, application of 3 mM GAS-NPs reduced the mite population average of 39%, while Cu-NPs caused 33.9% reduction percentage in the same concentration at 50% of water holding capacity. GAS-NPs gave the highest averages of plant height, branches, and leaves a number and fresh and dry weight of plant at 1 mM.

Moreover, Authors suggest that treatment with Cu-NPs or GAS-NPs at 50% of water holding capacity had influence on anatomical parameters, like increased the thickness of mid-vein and xylem and phloem tissues. Likewise, such treatment increased stem diameter due mainly to the increase in the thickness of cortex, phloem, and xylem tissues compared with the control.

The introduction part gave an sufficient background to the reader as well as the aim of the study was clearly underlined;

Materials and methods section was clearly and in detailed described in a repetitive way;

Results section needs substantial improvements and a lot of carefully conducted work:

-          Figure 1to 4 should be added in enlarged version, in current form is almost unreadable;

-          Figure 5 a and b -please correct (axis X on the graph & the legend);

-          Why number of insects in “normal condition” is lower than after drought stress treatment (after curcumin application)??

-          What kind of unit were used in figure ‘effect of nano curcumin and nanoglycyrrhizic acid under drought stress on anatomical characters of a soybean leaf’- TABLE 2 and TABLE 3;

-          Why “nano curcumin” and “Nanoglycyrrhizic” were writing in a different way (separate and/or capital letter) -please correct in the whole manuscript;

-          The same situation as in Table 3and 4- crude protein -what kind of unit, as well as total chlorophyll ??

-          Figure 7 should be correct at all, microphotographs are too small, with different orientation – upper epidermis should be on the top of the photo, therefore please rotate all in a appropriate way; furthermore, photos overlapped each other; Reader should clearly show the differences – phloem, xylem;

-          In the Figure 7 and 8 please add tissues markings- enlarge figure 8 the same as figure 7 to make the differences visible (!);

-          Table 2 and 3 did not show any changes described in discussion part ??- “as authors stated: Our findings show that nanoglycyrrhizic acid and nanocurcumin stimulate MAPK cascades, which improve plant growth by increasing photosynthesis pathways and growth hormones (Tables 2, 3).”- Please correct it;

-          Why in discussion a big part is about MAPK kinases and ROS condition??

-          Why some parts of discussion are in different fonts and font-size?

-          Why all tables in  the manuscript did not have statistical analyses??

In my opinion it is too early to publish presented manuscript in current form, because a lot of issues need to be corrected. Therefore, I encourage substantial improvement and resubmit manuscript in a corrected form;

Author Response

Response to reviewer 1

Greetings, I would like to acknowledge and appreciate your valuable comments.

Response to Review Report (Round 1)

Review comment 1: Figure 1to 4 should be added in enlarged version, in its current form is almost unreadable;

Response: Done, please check the manuscript

Review comment 2: Figure 5 a and b -please correct (axis X on the graph & the legend);

Response: Done, please check the manuscript

Review comment 3: Why the number of insects in “normal condition” is lower than after drought stress treatment (after curcumin application)??

Response: Please check the following references

Ximénez-Embún, M. G., Castañera, P., & Ortego, F. (2017). Drought stress in tomato increases the performance of adapted and non-adapted strains of Tetranychus urticae. Journal of insect physiology, 96, 73–81. https://doi.org/10.1016/j.jinsphys.2016.10.015.

Ximénez-Embún MG, Ortego F, Castañera P. Drought-Stressed Tomato Plants Trigger Bottom-Up Effects on the Invasive Tetranychus evansi. PLoS One. 2016 Jan 6;11(1):e0145275. doi: 10.1371/journal.pone.0145275. PMID: 26735490; PMCID: PMC4703393.

English-Loeb, G.M. Plant drought stress and outbreaks of spider mites: A field test. Ecology 1990, 71,

1401–1411.

- Reduction of mites in response to curcumin nanoparticles may be due to induction-specific signalling in plants; increasing phenolic compounds may also be factors in mite reduction.

Review comment 4: What kind of unit were used in figure ‘effect of nano curcumin and nanoglycyrrhizic acid under drought stress on anatomical characters of a soybean leaf’- TABLE 2 and TABLE 3;

Response: Done, please check the manuscript

Review comment 5: Why “nano curcumin” and “Nanoglycyrrhizic” were writing in a different way (separate and/or capital letter) -please correct in the whole manuscript;

Response: Done, please check the manuscript

Review comment 5: The same situation as in Table 3and 4- crude protein -what kind of unit, as well as total chlorophyll ??

Response: Done, please check the manuscript

Review comment 6: Figure 7 should be correct at all, microphotographs are too small, with different orientation – upper epidermis should be on the top of the photo, therefore please rotate all in a appropriate way; furthermore, photos overlapped each other; Reader should clearly show the differences – phloem, xylem;

Response: Done, please check the manuscript

Review comment 7: In the Figures 7 and 8 please add tissues markings- enlarge figure 8 the same as figure 7 to make the differences visible (!);

Response: Done, please check the manuscript

Review comment 8: Table 2 and 3 did not show any changes described in discussion part ??- “as authors stated: Our findings show that nanoglycyrrhizic acid and nanocurcumin stimulate MAPK cascades, which improve plant growth by increasing photosynthesis pathways and growth hormones (Tables 2, 3).”- Please correct it;

Response: Done

Discussion

Nanotechnology has been considered one of the most important technologies of the modern Era. Nanotechnology has been applied in agriculture with great impact in many agricultural disciplines [9]. For instance, natural nanomaterials can enhance plant growth, secondary metabolites, active compounds, resistance to various stimuli, crop yields, and distinct physiological pathways [23].

In this study, it is logic we found the T. urticae slightly attacking the drought stressed plant more than relaxed plant in normal conditions even the difference is not significant. However, the positive impact of nano- glycyrrhizic acid and nano-curcumin treatments is reduction in no. of insects significantly in both drought and normal conditions in both concentrations of Cu-NPs and GA-NPs (Table1, Fig.5). may be the increase of resistance is correlated with increasing in nitrogen, potassium, and phosphorus content, as well as an increase in photosynthetic pigments and capacity (table 2, 6), which is unfavorable for T. urticae [26, 27] and directly causes an observed reduction in its populations. current research studies proved that the application of nano-glycyrrhizic acid and nano-curcumin on the soybean (Glycine max) enhanced the photosynthesis rate by increasing Leaf area root length, water, and nutrient materials absorption, which impacted chlorophyll percentage, plant height, number of branches, fresh and dry weight, oil percentage and contents, and plant resistance against both biotic and abiotic stresses (Tables 2, 3 and Figure 6). The impcts of nanoparticles treatments in plant yields was reported before [24] when investigator clarified that a better growth of Solanum lycopersicum and high fruit yield due to TiO2 NPs foliar treatment. Also, the highly favorable effects of Ag NPs on grains/spike, 100-grain weight, and grain yield per pot in wheat [25].   Exogenous application of nano- glycyrrhizic acid and nano-curcumin to the soybean (Glycine max) may induce specific pathways that have a significant impact on plant growth, physiological activities, secondary metabolites, various pathways, and active compound production [4, 23].

Our findings show that nano- glycyrrhizic acid and nano-curcumin improve plant growth by increasing photosynthesis pathways and growth hormones (Tables 2, 3). many investigators explain this increase in plant growth due to the effect of nano- glycyrrhizic acid and nano-curcumin MAPKs pathway induction which regulate many important cellular processes such as cell division, different developmental processes regulated by hormones, stress responses, metabolism, and biologically active compounds [28] in addition to the role of MAPK cascades in stomata controlling, callose deposition, and seed germination [29, 30]. Other investigators suggest the positive effect of nano-glycyrrhizic acid and nano-curcumin on Oxidative signal Inducible1 gene (Oxi1) a protein kinase member of the AGC family that links ROS accumulation to plant response and resistance to various stimuli, as well as oxidative burst-mediated signaling in plant roots [31, 32].

The size of the midrib was further increased, as the percentages increased by 42.9% in length and 87.3% in width, referring to an increase in the dimensions of the vascular bunch of the midrib since it reached 50% of the length and 200% of the width, along with the thickness of the collenchyma above the bundle (40%) and below the vascular bundle (52.5%). The increase in midrib vascular bundle dimensions is due to 80% growth of the thickness of phloem tissue and 71.4% of xylem tissue.

On leaves with a thicker mesophyll rose, the lowest density of phytophagous mites occurred. Some species were more resistant to spider mites with thicker cuticles and epidermis. Furthermore, mites favored less the upper leaf surface, as they are thicker cuticles and epidermis than the lower surface [14, 33]. Within most often feed exclusively on cells within the mesophyll parenchyma; the disturbed cells adjacent to the leaf surface's epidermal layer from which the mites feed. As the thickness of leaves ranges from 100 to 150 μm, the style of the mites can completely cross a leaf and it can be reached by a palisade or by spongy mesophyll regardless of the surface of the leaf, however, the basis is currently unknown for this preference [34]. Although the treatment with nano-glycyrrhizic acid gives positive results in most traits, but we can say the nano-glycyrrhizic acid and nano-curcumin treatments are significantly improved most of soybean traits and tolerance to biotic and abiotic stress.

Review comment 9: Why in discussion a big part is about MAPK kinases and ROS condition??

Response: Please check the discussion part.

Review comment 10: Why some parts of discussion are in different fonts and font-size?

Response: Format corrected accordingly, please check the manuscript

Review comment 11: Why all tables in the manuscript did not have statistical analyses??

Response: Statistical analysis has been added.

Reviewer 2 Report

Dear Dr. Ramadan,

your manuscript titled "Influence of spraying nono-cucurmin ....in response to T. urticae infestation and drought stress" is interesting and applies innovative methods well suited for publication in Plants. Moreover, the problem addressed is very important under climate change conditions with increasing periods of drought in many countries. Pest control based on natural compounds needs to be enhanced, thus your study is a very useful contribution.

The current version of your manuscript, however, needs in-depth editing and language improvement.

In the following I will outline my comments:

Abstract

Many sentences seem to be not finished or are constructed in such a way that the text cannot be fully understood. Please re-write with care. 

Line 23-42 , a concluding sentence is missing

Introduction

line 47: esential is not the adequate word, please re-write

line 53-54 here you used 4 times in one sentence the word drought - change

Thoughout the language used makes it difficult to understand and evaluate the content. Please re-write.

Material and methods

give exact longitude, altitude etc. of your study location

cite methods used with corresponding publications of first use

Results

pictures of leaf anatomy are not numbered and placed partly on top of each other

Same for stem anatomy photos

Figure 5 a and b

Please explain in the figure legend the abbreviations used so that the figure can be understood by itself.

Same for table 1.

Discussion

Again, re-write with improved language grammar and word use. A last paragraph with conclusions and outlook as well as an exhaustive comparison with related studies is needed. 

Conclusion

One last sentence that summarizes your results is missing.

Statistical analyses need to be explained in more detail. The citation of a related study is not sufficient here to be able to evaluate your statistics.

Author Response

Response to Reviewer 2

Greetings, I would like to acknowledge and appreciate your valuable comments.

Response to Review Report (Round 1)

Review comment 1: Abstract, many sentences seem to be not finished or are constructed in such a way that the text cannot be fully understood. Please re-write with care.

Response: Please check the abstract

Modern nanotechnology has been credited as one of the most significant inventions of the 21th century. Many agricultural disciplines have been affected by nanotechnology in agriculture. Pest control based on natural compounds needs to be enhanced, also enhancing plant growth under climate change conditions with increasing period of drought in many countries is a very vital aim. Thus, the effect of curcumin nanoparticles (Cu-NPs) and glycyrrhizic acid nanoparticles (GA-NPs) as a foliar application under water deficit on natural infestation with the two-spotted spider mite Tetranychus urticae, plant growth and yield, anatomical and chemical parameters was investigated during this study. The obtained results revealed that drought stress over the two studied seasons significantly increased the population of T. urticae and decreased all morpholog-ical and yield characters.  Application of 3 mM GA-NPs reduced the mite population average of 39%, while using the same concentration of Cu-NPs caused 33.9% reduction percentage under drought stress. Using 1 mM GA-NPs gave the highest averages of plant height, number of branches and leaves/ plant fresh and dry weight. Moreover, the number of pods, 100 seed weight and seed yield (kg/ha) increased significantly as a result of spraying with GA-NPs under water deficit. From results, water deficit decreased the values of the leaf and stem anatomical parameters. Treatment with Cu-NPs or GA-NPs under drought stress increased the thickness of mid-vein, xylem, and phloem tissues. Likewise, such treatment increased stem diameter due mainly to the increase in the thickness of cortex, phloem, and xylem tissues compared with the control. Spraying plants with GA-NPs at 1mM increased the percentages of nitrogen, phosphor, and potassium in seeds in addition to total chlorophyll. Moreover, glutamate, aspartate, leucine, arginine, Lysine, glycine, tyrosine, tryptophan, and methionine concentrations did not differ significantly (P > 0.05) in response to all the studied levels of Cu-NPs or GA-NPs either under normal irrigation or drought condition. In light of these findings, researchers and producers should apply and test both Cu-NPs and GA-NP as nano-fertilizer natural sources on economi-cally viable crops.

Review comment 2: Line 23-42 , a concluding sentence is missing

Response: done, please check the abstract

Review comment 3: Introduction: line 47: essential is not the adequate word, please re-write

Response: Soybean (Glycine max (L.)) is a globally significant plant that is widely used to produce a variety of ingredients.

Review comment 4: line 53-54 here you used 4 times in one sentence the word drought – change

Thoughout the language used makes it difficult to understand and evaluate the content. Please re-write.

Response: Climate change is projected to have a major impact on precipitation, temperature, and potential evapotranspiration, affecting the frequency and severity of meteorological droughts. One important issue for assessing future impacts is the effect of changes in meteorological drought have an impact on soil water stress and hydrological droughts such as groundwater and stream water shortages. For instance, soil water drought is relevant to agriculture, land ecosystems, and health by heat waves, whereas the water resources (agriculture, domestic and industrial water use), aquatic, power generation, and sailing of water, among others, have significance in hydrological drought [1].

Review comment 5: Material and methods, give the exact longitude, altitude etc. of your study location.

Response: done, please check the manuscript.

Review comment 6: cite methods used with corresponding publications of first use

Response: done, please check the manuscript.

Review comment 7: Results, pictures of leaf anatomy are not numbered and placed partly on top of each other

Same for stem anatomy photos

Response: Done, please check the results section

Review comment 8: Figure 5 a and b, please explain in the figure legend the abbreviations used so that the figure can be understood by itself.

Response: done, please check the results section

Review comment 9: Same for table 1.

Response: done, please check the results section

Review comment 9: Discussion, Again, re-write with improved language grammar and word use. A last paragraph with conclusions and outlook as well as an exhaustive comparison with related studies is needed.

Response: done, please check the discussion part.

Discussion

Nanotechnology has been considered one of the most important technologies of the modern Era. Nanotechnology has been applied in agriculture with great impact in many agricultural disciplines [9]. For instance, natural nanomaterials can enhance plant growth, secondary metabolites, active compounds, resistance to various stimuli, crop yields, and distinct physiological pathways [23].

In this study, it is logic we found the T. urticae slightly attacking the drought stressed plant more than relaxed plant in normal conditions even the difference is not significant. However, the positive impact of nano- glycyrrhizic acid and nano-curcumin treatments is reduction in no. of insects significantly in both drought and normal conditions in both concentrations of Cu-NPs and GA-NPs (Table1, Fig.5). may be the increase of resistance is correlated with increasing in nitrogen, potassium, and phosphorus content, as well as an increase in photosynthetic pigments and capacity (table 2, 6), which is unfavorable for T. urticae [26, 27] and directly causes an observed reduction in its populations. current research studies proved that the application of nano-glycyrrhizic acid and nano-curcumin on the soybean (Glycine max) enhanced the photosynthesis rate by increasing Leaf area root length, water, and nutrient materials absorption, which impacted chlorophyll percentage, plant height, number of branches, fresh and dry weight, oil percentage and contents, and plant resistance against both biotic and abiotic stresses (Tables 2, 3 and Figure 6). The impcts of nanoparticles treatments in plant yields was reported before [24] when investigator clarified that a better growth of Solanum lycopersicum and high fruit yield due to TiO2 NPs foliar treatment. Also, the highly favorable effects of Ag NPs on grains/spike, 100-grain weight, and grain yield per pot in wheat [25].   Exogenous application of nano- glycyrrhizic acid and nano-curcumin to the soybean (Glycine max) may induce specific pathways that have a significant impact on plant growth, physiological activities, secondary metabolites, various pathways, and active compound production [4, 23].

Our findings show that nano- glycyrrhizic acid and nano-curcumin improve plant growth by increasing photosynthesis pathways and growth hormones (Tables 2, 3). many investigators explain this increase in plant growth due to the effect of nano- glycyrrhizic acid and nano-curcumin MAPKs pathway induction which regulate many important cellular processes such as cell division, different developmental processes regulated by hormones, stress responses, metabolism, and biologically active compounds [28] in addition to the role of MAPK cascades in stomata controlling, callose deposition, and seed germination [29, 30]. Other investigators suggest the positive effect of nano-glycyrrhizic acid and nano-curcumin on Oxidative signal Inducible1 gene (Oxi1) a protein kinase member of the AGC family that links ROS accumulation to plant response and resistance to various stimuli, as well as oxidative burst-mediated signaling in plant roots [31, 32].

The size of the midrib was further increased, as the percentages increased by 42.9% in length and 87.3% in width, referring to an increase in the dimensions of the vascular bunch of the midrib since it reached 50% of the length and 200% of the width, along with the thickness of the collenchyma above the bundle (40%) and below the vascular bundle (52.5%). The increase in midrib vascular bundle dimensions is due to 80% growth of the thickness of phloem tissue and 71.4% of xylem tissue.

On leaves with a thicker mesophyll rose, the lowest density of phytophagous mites occurred. Some species were more resistant to spider mites with thicker cuticles and epidermis. Furthermore, mites favored less the upper leaf surface, as they are thicker cuticles and epidermis than the lower surface [14, 33]. Within most often feed exclusively on cells within the mesophyll parenchyma; the disturbed cells adjacent to the leaf surface's epidermal layer from which the mites feed. As the thickness of leaves ranges from 100 to 150 μm, the style of the mites can completely cross a leaf and it can be reached by a palisade or by spongy mesophyll regardless of the surface of the leaf, however, the basis is currently unknown for this preference [34]. Although the treatment with nano-glycyrrhizic acid gives positive results in most traits, but we can say the nano-glycyrrhizic acid and nano-curcumin treatments are significantly improved most of soybean traits and tolerance to biotic and abiotic stress.

Review comment 10: Conclusion, One last sentence that summarizes your results is missing.

Response: done

Conclusion

Natural nanomaterials such as curcumin and glycyrrhizic acid, in both concentrations, improved soybean growth parameters. Curcumin nanoparticles and glycyrrhizic acid nanoparticles may both act as nano-fertilizer sources, mitigating the negative effects of water scarcity on soybean growth and yield. These treatments resulted in a significant decrease in T. urticae densities over the growing season, as well as a decrease in mite female fertility. The highest concentrations of curcumin and glycyrrhizic acid nanoparticles had nearly the same impact as lower concentrations, so low concentrations were recommended for economic impact applications. Both curcumin nanoparticles and glycyrrhizic acid ammonium salt nanoparticles may act as nano-fertilizer sources to alleviate the harmful effect of water deficiency on Soybean growth and its yield. In light of these findings, researchers and producers should apply and test nano-fertilizers on economically viable crops.

Review comment 11: Statistical analyses need to be explained in more detail. The citation of a related study is not sufficient here to be able to evaluate your statistics.

Response: done, please check the manuscript.

Round 2

Reviewer 1 Report

Dear Editors,

Authors significantly improved manuscript, espacially statistical analyses were at least added;

I have reservations to the low quality of presented and analysed anatomical studies, but I will be honest I don't want to block the manuscipt

sincerely

Author Response

Author's Reply to the Review Report (Reviewer 1)

Reviewer 1 comment: I have reservations about the low quality of presented and analysed anatomical studies, but I will be honest I don't want to block the manuscript.

Response to reviewer 1:

3.4. Anatomical studies

 3.4.1. Leaf anatomy

Interestingly, treatments with Cu-NPs and GA-NPs enhanced the anatomical characteristics of leaves (Tables 2 and Fig. 8). Under drought conditions (50% FC), anatomical characteristics of leaves enhanced significantly in response to 1mM GA-NPs. The recorded data were, 1125 µm midvein thick, 160 µm Lamina,  65 µm  plaside thickness , 210 µm Xylem tissue thick, 155 µm Phloem tissue thick and dimensions of the main vascular bundle of mid-vein (540 µm length -770 µm width).  However in 3 mM Cu-NPs , leaf midvein  was 945 µm, Lamina was 184 µm, palisade thickness  was 74 µm, Xylem tissue was195 µm, Phloem tissue (150 µm), and Dimensions of the main vascular bundle of mid-vein were  445µm length and 485 µm width.   Wihle control 50% FC data were 905 µm, 175 µm, 70 µm, 160 µm, 110 µm and  265 µm length -320 µm width, respectively. control 100% FC data were 930 µm, 165 µm, 68 µm, 177 µm, 135 µm and  385 µm length -405 µm width, respectively.

Figure 8. Transverse sections through the blade of soybean leaves at the age of 90 days, A. Control plant (100% FC), B. Negative control plant (50% FC), C. Plants sprayed with Cu-NPs at 3mM, and D. Plant sprayed with GA NPs at 1mM, 40X.

Table 2. The effect of Cu-NPs (3mM) and GAS-NPs (1mM) on anatomical characters of soybean leaf (µm) under drought stress (50% FC).

Treatments

Midvein thick

Lamina thick

Palisade thick

Spongy thick

Dimensions of the main vascular bundle of mid-vein

Xylem tissue thick

Phloem tissue thick

Length

Width

Control (100% FC)

930±41.84b

165±2.89b

68±0.95ab

82±6.13bc

385±2.75c

405±24.66c

177±8.94bc

135±2.00b

Drought (50% FC)

905±7.64b

175±3.59abc

70±2.04ab

90±1.64b

265±8.24d

320±7.67d

160±6.87c

110±6.47c

Cu-NPs (3 mM) + 50% FC

945±9.28b

184±4.10a

74±2.81a

105±3.14a

445±8.75b

485±10.77b

195±3.24ab

150±5.13a

GA-NPs (1 mM) + 50% FC

1125±11.64a

160±5.27b

65±3.06b

75±0.61c

540±10.12a

770±5.03a

210±4.04a

155±2.84a

The data represent the means ± standard error. Means followed by the same superscript letter in each column were not significantly different according to Duncan’s multiple range test (p ≤ 0.05).

3.4.2 Stem anatomy

Under drought conditions (50% FC), anatomical characteristics of the stem enhanced significantly in response to 1mM GA-NPs and recorded 3287µm Main stem diameter, 220 µm Cortex thickness, 22 vascular bundles, 240 µm phloem tissue thickness, 650 µm Xylem tissue thick and 1685 µm Parenchymatous pith thick respectively, compared to control (50% FC- 100% FC). Correspondingly, the treated plants with 3 mM CU-NPs enhanced anatomical characteristics of stem and scored 3177µm Main stem diameter, 165 µm Cortex thickness, 18 vascular bundles, 215 µm phloem tissue thickness, 590 µm Xylem tissue thick and 1620 µm Parenchymatous pith thick respectively, compared to control (50% FC- 100% FC). (Table 3).

Table 3. The effect of  Cu-NPs (3mM) and GAS-NPs (1mM) on anatomical characters of soybean stem (µm) under drought stress (50% FC).

Treatments

Main stem diameter

Cortex thickness

Number of vascular bundles

Phloem tissue thickness

Xylem tissue thick

Parenchymatous pith thick

Control (100% FC)

3020±38.14b

170±5.03b

12±0.40d

150±1.00c

320±1.15c

1555±7.23c

Drought (50% FC)

2190±51.52c

195±2.13a

15±0.38c

130±1.15d

255±1.53d

1330±4.51d

Cu-NPs (3 mM) + 50% FC

3177±61.42ab

165±4.41c

18±0.40b

215±3.06b

590±7.37b

1620±4.73b

GA-NPs (1 mM) + 50% FC

3287±43.40a

220±3.27a

22±0.70a

240±1.15a

650±7.55a

1685±4.36a

The data represent the means ± standard error. Means followed by the same superscript letter in each column were not significantly different according to Duncan’s multiple range test (p ≤ 0.05).

Results presented in table (3) and Figure (9) indicated that foliar spray with 3mM Cu-NPs increased the diameter of the main stem by 45.1%. The number of vascular bundles increased by 20.0 %. The increased thickness of phloem and xylem tissues and parenchymatous pith was 65.4, 131.4, and 21.8 %, respectively. While the thickness of the cortex decreased by 15.4% less than the control.  Moreover, the thickness of the main stem increased with the application of GA-NPs at 1 mM over control by 50.1%. The number of vascular bundles increased by 46.7 %. On the other hand, the thickness of cortex, phloem, xylem tissues, and parenchymatous pith was markedly increased under the same conditions by 12.8, 84.6, 154.9, and 26.7% respectively. On the other hand, water deficit (50% FC) caused significant decreases in all anatomical characters except cortex thickness which was increased by 14.7% and the number of vascular bundles was increased by 25% compared with the control. Whereas the main stem diameter decreased by 27.5% below control. The number of vascular bundles was increased by 25% compared with the control. While the thickness of phloem and xylem tissues and parenchymatous pith decreased by 13.3, 20.3, and 14.5% respectively, compared with the control (Table 3).

Figure 9. Transverse sections through the 4th internode of soybean stem at the age of 90 days, A: Control plants (100% FC), B: Control plants (50% FC), C:  Plants sprayed with Cu-NPs at 3mM, and D: Plants sprayed with GA-NPs at 1mM, 100X.
